# Pyroptosis of Macrophages Induced by *Clostridium perfringens* Beta-1 Toxin

**DOI:** 10.3390/toxins15060366

**Published:** 2023-05-29

**Authors:** Siyu Zhang, Lingling Ma, Fuyang Song, Dong Wang, Kesong Shi, Yong Li, Jin Zeng, Yujiong Wang

**Affiliations:** Key Laboratory of Ministry of Education for Conservation and Utilization of Special Biological Resources in the Western China, College of Life Science, Ningxia University, Yinchuan 750021, China; zhangsiyu1997@163.com (S.Z.); hz15202600308@163.com (L.M.); songfy26@163.com (F.S.); dongw1998@163.com (D.W.); sg1033425914@163.com (K.S.); liyong7732@nxu.edu.cn (Y.L.)

**Keywords:** *Clostridium perfringens* beta-1 toxin, pyroptosis, macrophages, Caspase-1/NLRP3 pathway

## Abstract

*Clostridium perfringens* beta-1 toxin (CPB1) is responsible for necrotizing enteritis and enterotoxemia. However, whether the release of host inflammatory factors caused by CPB1 is related to pyroptosis, an inflammatory form of programmed cell death, has not been reported. A construct expressing recombinant *Clostridium perfringens* beta-1 toxin (rCPB1) was created, and the cytotoxic activity of the purified rCPB1 toxin was assessed via CCK-8 assay. The rCPB1-induced macrophage pyroptosis by assessing changes to the expression of pyroptosis-related signal molecules and the pyroptosis pathway of macrophages using quantitative real-time PCR, immunoblotting, ELISA, immunofluorescence, and electron microscopic assays. The results showed that the intact rCPB1 protein was purified from an *E. coli* expression system, which exhibited moderate cytotoxicity on mouse mononuclear macrophage leukemia cells (RAW264.7), normal colon mucosal epithelial cells (NCM460), and human umbilical vein endothelial cells (HUVEC). rCPB1 could induce pyroptosis in macrophages and HUVEC cells, in part through the Caspase-1-dependent pathway. The rCPB1-induced pyroptosis of RAW264.7 cells could be blocked by inflammasome inhibitor MCC950. These results demonstrated that rCPB1 treatment of macrophages promoted the assembly of NLRP3 inflammasomes and activated Caspase 1; the activated Caspase 1 caused gasdermin D to form plasma membrane pores, leading to the release of inflammatory factors IL-18 and IL-1β, resulting in macrophage pyroptosis. NLRP3 may be a potential therapeutic target for *Clostridium perfringes* disease. This study provided a novel insight into the pathogenesis of CPB1.

## 1. Introduction

*Clostridium perfringens* is a kind of anaerobic Gram-positive bacterium that widely exists in the intestinal tissue of humans and animals. It is one of the main pathogenic bacteria that induce necrotizing enteritis and enterotoxamia, which pose a serious threat to animal husbandry production [1]. The bacterium can ferment sugars such as glucose, lactose, and sucrose, and produce large amounts of gas within tissues, leading to it being named perfringens, which is from the Latin for through (per) and burst (frango). In 1891, during an autopsy on a 38-year-old man, William H. Welch was the first to isolate *Clostridium perfringens* and report on its characteristics [2]. The virulence of *Clostridium perfringens* is largely attributable to its ability to produce toxins. At present, 20 types of exotoxins and enzymes produced by *Clostridium perfringens* have been identified [3]. In early investigations, *Clostridium perfringens* was divided into five types according to the four toxins produced [4]. In 1892, Welchii et al. isolated *Clostridium perfringens* type A from decayed cadavers, followed by types B (1926), C (1929), D (1932), and E (1943). With further investigation, *Clostridium perfringens* enterotoxin (CPE) and necrotizing enterotoxin B (NetB) were found to play important roles in diseases caused by *Clostridium perfringens*, so a new classification scheme needed to be developed. In 2018, two new types of *Clostridium perfringens* were added, namely, type F and type G, and thus the classification of the bacterium was extended to seven types (A to G) [5]. At present, *Clostridium perfringens* is known to produce α (CPA), β1 (CPB1), ε (ETX), ι (ITX), CPE, Perfringolysin O (PFO), β2 (CPB2), NetB, Kappa, Delta, and other kinds of toxin [3]. Among them, CPA, CPB, ETX, ITX, CPE, and NetB are the most influential and lethal toxins and can cause tissue, nervous system, and intestinal toxicity.

Studies have shown that *Clostridium perfringens* beta-1 toxin (CPB1) is encoded by the plasmid-encoded gene *cpb1*. The gene *cpb1* is about 1.0 Kb in length, and encodes CPB1 protein with a molecular weight of approximately 34 kDa [6]. CPB1 is produced by *Clostridium perfringens* types B and C, of which type C is a highly destructive pathogen [7]. The disease known as Darmbrand [8] appeared after the World War II, and was determined to be necrotizing enteritis caused by *Clostridia perfringens* type C infection; this disease was also an important reason for the high mortality rate of children in Papua New Guinea [9]. CPB1 is an important virulence factor for hemorrhagic intestinal necrosis induced by *Clostridium perfringens* type C [7], mainly occurring in neonates of many animal species (especially in newborn animals) and human beings [10].

As a form of programmed cell death, pyroptosis has attracted increasing attention in recent years because of the occurrence and development of diseases, and has become a hotspot in the field of life science [11]. There are many types of exogenous stimuli that can cause pyroptosis, in the canonical inflammasome activation pathway, [12] inflammasomes [13] such as NLR Family Pyrin Domain Containing 3 (NLRP3), which can sense bacteria, viruses, toxins, and ATP stimuli [14]. These stimuli indirectly activate NLRP3 through K^+^ outflow and lead to the binding of NIMA-related kinase 7 to NLRP3, thereby triggering the oligomerization of NLRP3. NLRP3 is then activated by an apoptosis-associated speck-like protein containing a CARD, which in turn activates cysteinyl aspartate specific proteinase 1 (Caspase 1) to produce interleukin (IL)-18 and IL-1β, and shear gasdermin D (GSDMD) to release the GSDMD-N domain, a peptide with the ability to form membrane pores, ultimately resulting in the secretion of activated IL-18 and IL-1β [15]. This signaling cascade of the NLRP3 and Caspase-1-dependent activation of pyroptosis is called the canonical pathway. Since 1995, a number of studies have found that Caspase 11 in mouse, as well as Caspase 45 in human cells [16,17], can induce the death of macrophages in response to a variety of Gram-negative bacterial infections, and the result is similar to the pyroptosis induced by Caspase 1 in morphology. This is called the non-canonical pathway [18]. Both pyroptosis and apoptosis were found to result in nuclear enrichment and chromosome DNA breakage [19,20], and TUNEL and Annexin V staining were positive. Differing from apoptosis, pyroptosis can lead to the formation of membrane pores and the destruction of membrane integrity, resulting in the release of cell contents, an increase in permeability, a release of inflammatory factors, and thus the induction of an inflammatory response.

Currently, accumulating studies have shown that, after infection or injury, K^+^ outflow activates the NLRP3 inflammasome, which leads to pyroptosis. As a perforated toxin, CPB1 can affect the membrane permeability of THP-1 cells. With the increase in CPB1 concentration and prolonged action, the release of K^+^ also increases. When the toxin was subcutaneously injected, it caused the rapid exudation of plasma and a significant increase in IL-1β and TNF-α at the injection site. Based on the above findings, we aimed to ascertain whether CPB1 induces pyroptosis by activating the inflammasome of stimulated cells through the Caspase-1-dependent pathway or the Caspase-4/5/11 pathway. We also attempted to investigate whether the release of inflammatory factors induced by CPB1 caused pyroptosis. Therefore, pyroptosis of macrophages induced by CPB1 was explored, and the findings will provide new insights into the pathogenic mechanisms of CPB1.

## 2. Results

### 2.1. Expression and Purification of Recombinant CPB1 (rCPB1) Toxin

First, the target gene in the recombinant plasmid was identified, and a gene with a size of approximately 1000 bp was amplified by PCR (Figure 1a). The result of *Eco*R I/*Hin*d III and *Eco*R I/*Not* I digestion identified the expected DNA bands of approximately 500 bp and 1000 bp (Figure 1a). The recombinant plasmid of the correct identification was sequenced by nucleotide sequencing, and the NCBI BLAST results showed an identical sequence to that reported in GenBank X83275.1 (Figure 1b). The results indicated that the *cpb1* gene was successfully inserted into the pTIG-Trx expression vector, and the recombinant plasmid was successfully constructed. The plasmid was named pTIG-Trx-*cpb1*, and the strain was named BpTc1. Then, the target protein was obtained by the denaturation and renaturation of inclusion bodies, and the results of SDS-PAGE and Western blot suggested that rCPB1 protein was successfully purified (Figure 1c,d). The molecular weight of the 34 kDa band of the protein of interest shown in SDS-PAGE, and the amino acid sequence analyzed using mass spectrum analysis (Jingjie Biotechnology Co., Ltd., Hangzhou, China) further confirmed that the rCPB1 protein expressed by the BpTc1 strain was the protein of interest (Figure 1e). As the identification of toxin activity was key to subsequent experiments, RAW264.7, NCM460, and HUVEC cells were treated with rCPB1 at different concentrations and times to determine toxicity. The results showed that rCPB1 could induce the death of the cells in a dose-dependent manner, and the concentrations causing 50% cell death of RAW264.7, NCM460, HUVEC cells were about 30 μg/mL, 30 μg/mL, and 20 μg/mL, respectively (Figure 2a–f). These results suggested that the rCPB1 protein with cytotoxicity was successfully prepared.

### 2.2. rCPB1 Induces Pyroptosis in RAW264.7 Cells

The above results showed that rCPB1 could induce the death of RAW264.7 cells. To investigate whether pyroptosis was involved in the cell death, RAW264.7 cells were treated with the ID50 concentration of rCPB1. The qRT-PCR assays revealed that the mRNA levels of NLRP3, Caspase 1, gasdermin D, IL-18, and IL-1β were significantly increased in a time-dependent manner (Figure 3a). A Western blot assay showed that the protein expression levels of NLRP3, Caspase 1, cleaved Caspase 1, gasdermin D, IL-18, and cleaved IL-1β were effectively increased, which was consistent with mRNA levels (Figure 3b). In addition, an ELISA assay demonstrated that the protein levels of IL-1β and IL-18 released to the culture medium were also increased (Figure 3c). Thus, these data clearly suggested that the pyroptosis of RAW264.7 cells could be induced by rCPB1.

### 2.3. rCPB1 Induces Pyroptosis in THP-1 Macrophage Cells

To explore whether the rCPB1 treatment of THP-1 macrophages cells can cause pyroptosis, a final concentration of 50 ng/mL of PMA was added to induce THP-1 cells to differentiate into THP-1 macrophages. The changes to the cell morphology were observed under transmission electron microscope, and the results showed cytoplasmic swelling and membrane rupture in the rCPB1-treatment group (Figure 4a). Assays of qRT-PCR, Western blotting and ELISA showed that the expression of pyroptosis-related cytokines was significantly up-regulated at both mRNA and protein levels after the rCPB1 treatment in a time-dependent manner (Figure 4b–d). These results suggested that rCPB1 could induce THP-1 macrophage pyroptosis.

### 2.4. rCPB1 Induces Pyroptosis in Bone Marrow-Derived Macrophages

According to the above experimental results, rCPB1 could induce pyroptosis in RAW264.7 and THP-1 macrophages. To better study the functions of pathogen-induced macrophages in the immune mechanism, primary cell lines are often used and bone marrow-derived macrophages (BMDM) are an excellent model. The morphology results and immunofluorescence assays showed that cells induced by the supernatant of the L929 cell culture had a phagocytic function and a large number of cell surface antigens CD11c and CD16 (Figure 5a), laying the foundation for subsequent experiments. Western blot assays and ELISA assays showed that the protein expression levels of pyroptosis-related cytokines were significantly up-regulated after rCPB1 treatment in a time-dependent manner (Figure 5b,c). These results indicated that rCPB1 induced BMDM pyroptosis.

### 2.5. rCPB1 Induces Pyroptosis in Other Cell Lines

To explore whether rCPB1 can cause pyroptosis in other cell lines besides macrophages, epithelial cells (NCM460) and endothelial cells (HUVEC) were treated with rCPB1 for different periods. A Western blot assay showed that, compared with the control group, the expression of pyroptosis-related proteins in HUVEC cells treated with rCPB1 were significantly up-regulated in a time-dependent manner (Figure 6a). However, it was not the case in NCM460 cells; that is, the expressions of pyroptosis-related proteins in NCM460 cells treated with rCPB1 did not change at any tested time points (Figure 6b). These results clearly indicated that rCPB1 might induce pyroptosis in macrophages, epithelial cells (NCM460), and endothelial cells (HUVEC) in a cell-context-dependent manner.

### 2.6. rCPB1 May Induce Macrophage Pyroptosis through the Typical Caspase-1-Dependent Pathway

To explore potential mechanisms underlying rCPB1-induced pyroptotic cell death in macrophages cells, three siRNA-Caspase 1 were, respectively, transfected into RAW264.7 cells. The qRT-PCR and Western blot assays revealed that siRNA-Caspase 1 significantly suppressed the mRNA and protein expression levels of Caspase 1, and the interference effect of siRNA-Caspase 1-1 was the most effective, with an interference effect of about 76% (Figure 7a). Therefore, siRNA-Caspase 1-1 was used in subsequent tests. A transmission electron microscopy assay showed less cytoplasmic swelling and membrane rupture in Caspase-1-silenced cells treated with rCPB1, suggesting that the inhibition of Caspase 1 prevented rCPB1-induced pyroptotic cell death in RAW264.7 cells (Figure 7b). Immunofluorescence assays showed that the expression of IL-1β in macrophages was significantly up-regulated after rCPB1 treatment, while siRNA-Caspase 1 effectively reduced its expression (Figure 7c). The qRT-PCR, Western blot, and ELISA assays further corroborated that silencing Caspase 1 suppressed the expression of pyroptosis-related factors (Figure 7d–f). These results suggested that the rCPB1-induced pyroptosis of RAW264.7 cells was mainly mediated by the Caspase-1-dependent pathway.

### 2.7. MCC950 May Be a Therapeutic Target for Caspase-1-Dependent Pyroptosis Induced by rCPB1

As the inhibitor of NLRP3, MCC950 has a promising applicability for NLRP3-related diseases. We next sought to examine whether MCC950 could inhibit rCPB1-induced pyroptosis by inhibiting NLRP3 and down-regulating the expression of pyroptosis-related factors. Western blot assays showed that the optimal inhibitory concentration of MCC950 was 2 μM upon rCPB1 treatment (Figure 8a). The Western blot assay and immunofluorescence assay demonstrated that rCPB1 treatment increased the expression of Caspase-1-dependent pyroptosis-related factors, but the addition of MCC950 significantly reduced its expression in response to rCPB1 treatment (Figure 8b,c). In addition, scanning electron microscopy results showed that rCPB1 treatment increased the number of cells with morphology of pyroptosis, while MCC950 reduced pyroptosis induced by rCPB1 in RAW264.7 cells (Figure 8d). These results suggested that MCC950 could inhibit the rCPB1-induced pyroptosis of RAW264.7 cells as a therapeutic target.

## 3. Discussion

Since 1965, extensive studies have been carried out to clarify the significance and mechanisms of programmed cell death. It has now been established that cells can undergo programmed death during development due to pathogen invasion or cell stress and metabolic disorders. Three programmed cell death pathways have been studied in depth in recent years: apoptosis, autophagy, and pyroptosis [21]. Apoptosis and autophagy do not produce an inflammatory response, while pyroptosis does [22,23]. Pyroptosis is a “double-edged sword” within the anti-infection immune processes, related to the inflammatory mechanism of the host. On the one hand, pyroptosis can activate the internal mechanism of cell death, release inflammatory factors, clear pathogens, and prevent infection. On the other hand, excessive Caspase 1 activation can lead to pathological inflammation. Although the onset of inflammation has a protective effect, excessive inflammation is harmful.

Pyroptosis pathogenesis can be divided into two pathways. Pyroptosis mediated by Caspase 1 and activated by inflammasomes is called the canonical pathway, while that mediated by Caspase-4/5 (in humans) or Caspase-11 (in mice) is called the non-canonical pathway. Pyroptosis can lead to the release of cellular contents, included damage-associated molecular patterns (DAMPs) and inflammatory factors (IL-1β and IL-18), causing inflammation [24].

*Clostridium perfringens* (*C. perfringens*) is a bacterium that exists widely in the environment, and also presents as normal intestinal flora in humans and domestic animals. *C. perfringens* can infect animals and humans by the ingestion of spores or through changes in the gut flora, followed by excessive growth of bacteria and toxin production, which causes enterotoxemia, necrotic hemorrhagic enteritis, and enterocolitis [25].

*C. perfringens* beta1 toxin (CPB1) is a lethal toxin secreted by types B and C. It is able to form membrane pores on susceptible cells, leading to cell distension and lysis. As a perforating toxin, CPB1 can affect the membrane permeability of THP-1 cells. With an increase in its concentration and the prolongation of the action time, the release of K^+^ in the cells will also increase. Interestingly, several studies have reported that the K^+^ efflux or Ca^2+^ flux from cells can induce NLRP3 inflammasome activation and assembly. It is well known that the assembly of the NLRP3 activates Caspase 1, and Caspase 1 activation leads to cytokine secretion. In some circumstances, it results in pyroptosis. Therefore, we speculate that CPB1 toxin maybe induce pyroptosis by causing K^+^ efflux from cells.

It has been reported in the literature that CPB1-toxin-induced mouse skin TNF-α and IL-1β levels increased in a dose-dependent manner, along with increased plasma extravasation. In the present study, we found that the expression levels of NLRP3, Caspase 1, Gasdermin D, IL-18, and IL-1β were up-regulated after RAW264.7, THP-1 macrophages, and BMDM were treated with rCPB1. The morphological changes of THP-1 macrophages were observed with an electron microscope, and it was found that the cell membrane ruptured and the cell contents overflowed in cells treated with rCPB1. Similarly, the rCPB1 toxin could cause the HUVEC cells to release inflammatory factors. These phenomena indicate that rCPB1 could induce pyroptosis in macrophage cells and endothelial cells. However, the results of qRT-PCR and Western blotting showed no significant changes in the mRNA and protein-related pyroptosis in NCM460 cells treated with rCPB1 toxin. The ELISA results showed the expression of IL-18 and IL-1β levels, and for the NCM460 cells there was no significant change either. This result suggests that the rCPB1 toxin protein can cause the death of colonic mucosal epithelium, but does not induce pyroptosis in NCM460 cells.

It has been reported that the CPB1 toxin can cause necrotizing enteritis in animals, which can lead to hemorrhagic necrosis of the intestinal mucosa. It is generally considered that the intestinal mucosal epithelium is the main target of CPB1-toxin-induced damage. However, studies have also shown that CPB1 does not cause direct damage to the intestinal epithelial cells of pigs after many relevant studies, although intestinal epithelial cells secrete low levels of IL-1β. A large number of immune cells, including macrophages and monocytes, are distributed in the intestinal epithelium and lamina propria, to compose the complex intestinal immune microenvironment. Several lines of study have found that the CPB1 toxin can induce different cytotoxic responses on different cell lines. The reason for this phenomenon may be related to the receptors on the plasma membrane surface of different cell types. For example, Takehara et al. discovered that the P2X purinoreceptor 7 (P2RX7) may be the receptor with the CPB1 toxin [26]. The P2RX7 receptor is widely expressed in many types of cells, including intestinal epithelial cells, and whether the CPB1 toxin caused the death of intestinal epithelial cells is directly or indirectly caused by the pore-forming toxin. Combined with the results of this experiment, the expression of pyroptosis-related factors did not occur in intestinal epithelial cells after being stimulated by CPB1 toxin. It is speculated that the death mode of intestinal epithelial cells may not be inflammatory death, such as pyroptosis. The release of inflammatory factors in animals may be caused by immune cells in the lamina propria of the intestinal tissue.

After considering our experimental results, we speculate that rCPB1 can induce the assembly of the NLRP3 inflammasome in macrophages, possibly through a K^+^ flow mechanism, which then activates Caspase 1. The activation of Caspase 1 in turn leads to the cleavage of GSDMD, and subsequently the formation of channels in the plasma membrane, and the release of the inflammatory cytokines IL-18 and IL-1β. Ultimately, a large number of inflammatory cytokines are produced through the inflammatory cascade, which destroys the intestinal epithelium, leading to diseases such as necrotizing enteritis in the host.

In this experiment, we also found that small-molecule inhibitors MCC950 and small interfering RNA in Caspase 1 could inhibit the occurrence of pyroptosis and alleviate the excessive release of inflammatory factors from macrophages. The NLRP3 inflammasome inhibitor MCC950 has been used in the treatment of some clinical diseases. In the treatment of necrotizing enteritis caused by *Clostridium perfringens*, whether it can be considered to use drugs that inhibit the excessive release of inflammatory factors by immune cells to avoid the subsequent inflammatory factor storm is still in question.

## 4. Conclusions

In conclusion, this study lays a foundation for the in-depth exploration of the damage mechanism of *C. perfringens* Beta1 toxin on different types of target cells, and also provides an insight into the pathogenic mechanism of CPB1 toxin.

## 5. Materials and Methods

### 5.1. Bacterial Strains, Plasmids, Cell Lines, Reagents, and Mice

*E. coli* BL21 (DE3) (pET30a-GFP) with green fluorescence were stored in our laboratory. BL21 (DE3) were purchased from Quanshijin Biotechnology Co., Ltd., Beijing, China. Mouse mononuclear macrophage leukemia cells (RAW264.7), human mononuclear leukemia cells (THP-1), normal colon mucosal epithelial cells (NCM460), and mouse fibroblast cell line L929 were purchased from the National Collection of Authenticated Cell Cultures; human umbilical vein endothelial cells (HUVEC) were donated by Professor Gao Liyang. The pTIG-Trx plasmid was donated by Professor Wang Jinglin. C57BL/6J mice were purchased from Weitong Lihua Experimental Animal Technology Co., Ltd., Beijing, China. 

### 5.2. Constructing the Plasmid That Expresses rCPB1

According to the *cpb1* DNA sequence in GenBank (X83275.1), the following primers were designed to amplify the encoding regions of the *cpb1* gene: cpb1-F: 5′ CGGAATTCCATATGGATATAGGCAAAACTACTACTAT′ and cpb1-R: 5′ GAATGCGGCCGCAATAGCTGTTACTTTATG′. Enzyme digestion was used to analyze the *cpb1* gene loci, and we found two *Hin*d III enzyme loci of approximately 500 bp and 600 bp; therefore, the recombinant plasmid was treated with *Eco*R I single-enzyme digestion and *Eco*R I/*Hin*d III and *Eco*R I/*Not* I double-enzyme digestion and subjected to agarose gel electrophoresis analysis. The amplified DNA fragment was cloned into the pTIG-Trx vector to construct the pTIG-Trx-*cpb1* plasmid for sequencing and transformed into BL21 (DE3) to induce the expression of CPB1.

### 5.3. Induction of the Expression of rCPB1 

BL21(DE3) pTIG-Trx-*cpb1* were isolated and cultured in LB medium with 50 μg/mL of ampicillin and 0.5 mM isopropyl β-D-1-thiogalactopyranoside (IPTG) in a 20 °C incubator with shaking overnight. Then, the bacterial protein samples were analyzed for expression of the rCPB1 on 12% SDS-PAGE.

### 5.4. Purification and Assay of rCPB1 

Given that the rCPB1 expressed from the pTIG-Trx vector had low solubility, it could be purified using inclusion body renaturation. In brief, the IPTG-induced BL21 (DE3) pTIG-Trx-*cpb1* were harvested with centrifugation (12,000 rpm, 10 min). The pellet was resuspended with PBS and ultrasonically broken. Then, the mixture was separated by centrifugation at 12,000 rpm for 30 min. The pellet was resuspended, respectively, in eluent I (50 mM Tris-Cl [pH 8.0], 10 mM EDTA, 5% glycerin, 0.1% TritonX-100, 100 mM β-mercaptoethanol), eluent II (eluent I with 0.3 M urea), or eluent III (eluent I with 0.6 M urea). To the bacteria obtained from the 1 L culture, 40 mL washing solution was added; centrifugation at 12,000 rpm for 15 min was conducted after each wash. The pellet was resuspended in denaturation solution (50 mM Tris-Cl [pH 8.0], 5% glycerin, 8 M urea) with ultrasound for 10 min. At this point, insoluble matter was significantly reduced. To bacteria obtained from 1 L culture, 20 mL of denaturation solution was added; the mixture was centrifuged at 12,000 rpm for 20 min. The target protein was now dissolved in the supernatant. We used a 3.5-kDa dialysis bag for renaturation of the supernatant in a low-temperature refrigerator. The process was as follows: addition of refolding solution I (50 mM Tris-Cl [pH 8.0], 5% glycerol, 6 M urea), dialysis for 2 h; addition of refolding solution II (50 mM Tris-Cl [pH 8.0], 5% glycerol, 4 M urea), dialysis for 3 h; addition of refolding solution III (50 mM Tris-Cl [pH 8.0], 5% glycerol, 2 M urea), dialysis for 2 h; and addition of refolding solution IV (50 mM Tris-Cl [pH 8.0], 5% glycerol) for 2 h or overnight. After renaturation, the solution was slightly turbid, but no obvious floccules appeared. The solution was centrifuged at 12,000 rpm for 20 min, and the supernatant containing the target protein was decanted. The endotoxin of the rCPB1 proteins was removed using a ToxinEraser™ Kit (GenScript, Piscataway, NJ, USA). The protein fractions were determined with 12% SDS-PAGE and an immunoblotting assay using a CPB1 antibody (Gene Tex GTX41667). Immune complexes were detected using the automatic chemiluminescence imager (GE Amersham Imager 600).

### 5.5. Cell Culture and Treatment

All cells were cultured in an incubator (37 °C, 5% CO_2_). RAW264.7 and HUVEC cells were cultured in DMEM medium. THP-1 and NCM460 cells were cultured in RPMI 1640 medium. Complete medium was composed of basal medium and 10% fetal bovine serum. Additionally, the cells in good condition were transferred to a six-well plate at a density of 1 × 10^6^ cells/mL and cultured overnight at 37 °C. The dose of rCPB1 that caused 50% cell death was added to the cells for treatment for 1 h, 2 h, 4 h, or 6 h. In the positive control group, 5 mM ATP (Sigma-Aldrich, St. Louis, MO, USA) was added for treatment: 0.5 h upon 1 μg/mL LPS treatment 5.5 h. In the inhibitor group, different doses of MCC950 were added for 1 h before rCPB1 treatment. 

PMA (Sigma-Aldrich, St. Louis, MO, USA), LPS (Sigma-Aldrich, St. Louis, MO, USA) and MCC950 (MCE, Monmouth Junction, NJ, USA) powders were dissolved in DMSO (Sigma-Aldrich, St. Louis, MO, USA) to achieve a stock solution, and kept in aliquot at −20 °C.

### 5.6. Cytotoxicity Assay

RAW264.7, NCM460, and HUVEC cells were used to evaluate the activity of the rCPB1. Cells were seeded into 96-well plates at a density of 8 × 10^4^ cells/mL per well for overnight culture, and six duplicate wells were set up in each group. The medium was changed for fresh medium, and different concentrations of toxin proteins were used for further incubation for different processing times. Then, CCK-8 solution was added to each well. After 2 h, the absorbance value was measured at 450 nm using a luciferase plate analyzer and the data were recorded.

### 5.7. Preparation and Phagocytosis of BMDM

Mice with 6–9 weeks of cervical dislocation were disinfected and the femurs removed. The bone marrow of the femurs was forced out with PBS until the wash was colorless. The supernatant was discarded after centrifugation (800 rpm, 5 min). The pellet was resuspended in RBC lysis buffer at 20s, followed by the same amount of DMEM complete medium, and centrifuged. The cells were resuspended in complete medium containing 20% L929 culture supernatant, and the solution was transferred to the six-well plate at a density of 1 × 10^6^ cells/mL. After culturing for 7 days, the induced *E. coli* BL21(DE3) (pET30a-GFP) were added to a final concentration of 5 × 10^7^ cells/mL and cultured at 37 °C for 2 h. The phagocytosis effect was analyzed and the images were captured.

### 5.8. Transfection

After being treated with rCPB1 with different times, the cells were washed 2–3 times with PBS. Lipofectamine™ iMAX transfection reagent (Thermo, Waltham, MA, USA) performed cell transfection through the concentration of siRNAs, according to the manufacturer’s recommendations.

The sequence for siRNAs is shown below:siRNA-Caspase1-1: 5′-GGTATCCAGGAGGGAATATGT-3′,siRNA-Caspase1-2: 5′-GCCCAAGGTGATCATTATTCA-3′,siRNA-Caspase1-3: 5′-GGCATTAAGAAGGCCCATATA-3′.

### 5.9. Real-Time Fluorescence Quantitative PCR

After being treated with rCPB1 for different times, the cells were washed 2–3 times with PBS and lysed with Trizol reagent (Ambion, Austin, TX, USA), so that total RNAs were extracted. The RNA concentration was measured using a Nanodrop 8000 nucleic acid protein analyzer, and the measured data were recorded. Then, complementary DNA was prepared by using cDNA Synthesis Kits (Thermo, Waltham, MA, USA). SYBR Green (QIAGEN, DUS, Hilden, Germany)-based quantitative PCR was performed by using the QuantStudio™ Detection System (Thermo, Waltham, MA, USA), according to the manufacturer’s recommendations. It counted according to 2^−ΔΔCT^ methods. Additionally, primers used in the qPCR assays are shown in Appendix A.

### 5.10. Enzyme-Linked Immunosorbent Assay

After being treated with rCPB1 for different times, culture media were collected and centrifuged to remove any cell debris or suspended cells. IL-1β and IL-18 released in the culture supernatant were measured by using a human or mouse ELISA kit (Lianshuo Biological Technology, Shanghai, China), according to the manufacturer’s recommendations.

### 5.11. Immunofluorescent Staining

After being treated with rCPB1 for different times, the cells were washed with PBS and fixed with 4% paraformaldehyde (RT, 20 min), then permeabilized with 0.5% Triton X-100 (RT, 20 min). Cells were blocked with 3% bovine serum albumin (RT, 1 h) and then incubated with primary antibodies at 4 °C overnight. After being washed, the cells were incubated with the secondary antibody (RT, 1 h) and mounted for fluorescence with DAPI (Thermo, Waltham, MA, USA). 

Among them, antibodies against NLRP3, Caspase 1, Actin, IL-18 and Cleaved IL-1β were purchased from Cell Signaling Technology, Danvers, MA, USA. Additionally, antibodies against Gasdermin D, GAPDH and HRP goat anti-mouse or goat anti-rabbit were purchased from Proteintech, Wuhan, China. The FITC-labeled CD11C antibody and Texas Red-labeled CD16 antibody were purchased from Thermo, Waltham, MA, USA.

### 5.12. Transmission Electron Microscope Observation

After being treated with rCPB1 for different times, the cells were gently collected in a centrifuge tube, centrifuged (800 rpm, 5 min), and the supernatant discarded. They were then washed twice with PBS and mixed with 2% glutaraldehyde fixative solution for 2 h at 4 °C; the solution was changed after 2 h, and this was repeated 3 times. The cells were then fixed with 1% hungry acid for 2 h, then rinsed with PBS for 5 min three times. They were treated at 4 °C with 30% ethanol, 50% ethanol, and 70% ethanol successively for 10 min, and treated at room temperature with 80% ethanol, 90% ethanol, and 100% ethanol for 10 min. Then, epoxy propane was added for 15 min and the process was repeated. Incomplete embedding solution and epoxy propane at ratios of 1:1 and 2:1 were added for 1 h, respectively, followed by overnight treatment with incomplete embedding solution, and incubation at 35 °C, 42 °C, and 60 °C for 6 h, 12 h, and 48h, respectively. The embedded cells were cut into slices, placed on slides, and observed using transmission electron microscopy.

### 5.13. Statistical Analysis

All data were obtained after at least three independent experiments. One-way ANOVA in GraphPad Prism 7.0 software was used for statistical analysis, and statistical evaluation of the differences was performed with a *t*-test for two groups. A difference was considered to be statistically significant at a *p* value of <0.05. Data are presented as the mean ± standard deviation (SD).

## Figures and Tables

**Figure 1 toxins-15-00366-f001:**
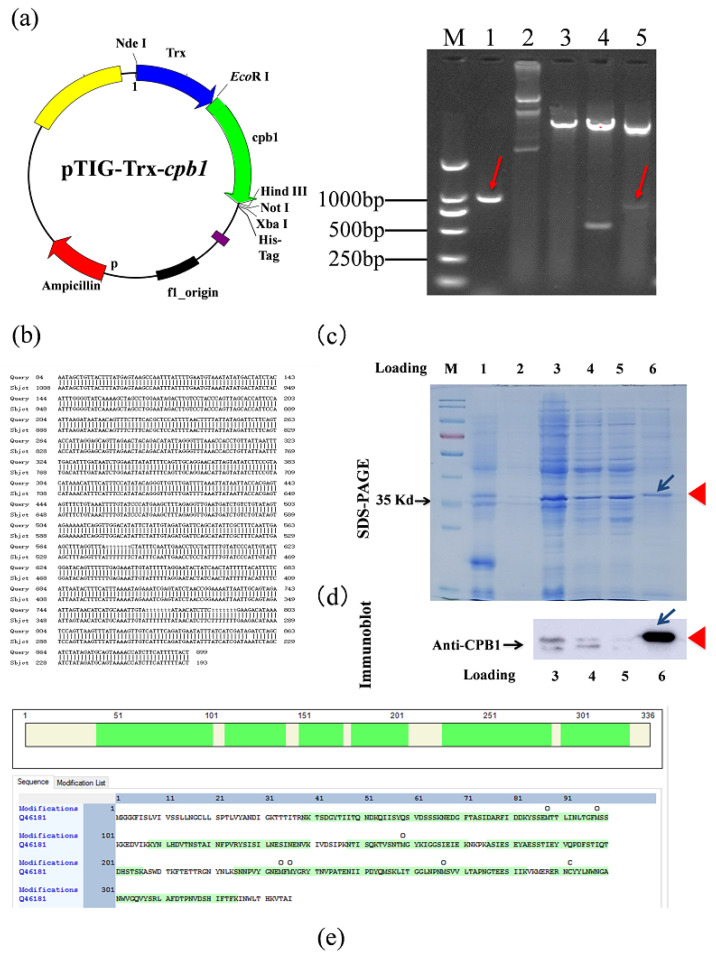
Preparation of recombinant toxin beta-1 of *Clostridium perfringens* (rCPB1). The expression of rCPB1 was induced by IPTG and lysed via sonication, and the target protein was obtained through inclusion body renaturation. (**a**) The map of recombinant plasmid, and an image of the nucleic acid gel electrophoresis results. Lane M, Marker. Lane 1, PCR fragment of *cpb1* gene. Lane 2, pTIG-Trx-*cpb1* plasmid. Lane 3, pTIG-Trx-*cpb1* plasmid digested by *Eco*R I. Lane 4, pTIG-Trx-*cpb1* plasmid digested by *Eco*R I/*Hin*d III. Lane 5, pTIG-Trx-*cpb1* plasmid digested by *Eco*R I/*Not* I. (**b**) The analysis of nucleotide sequence alignment. (**c**) An image of the results of SDS-PAGE (12%) analysis. Lane M, Maker. Lane 1, a lysate of BL21(DE3) transformed with plasmid pTIG-Trx. Lane 2, a lysate of recombinant strain. Lane 3, the lysate of recombinant strain was induced. Lane 4, the precipitation after washing in eluent I. Lane 5, the precipitation after washing in eluent II. Lane 6, rCPB1 protein. A band of protein of interest with expected MW of ∼34 kDa was observed in lanes 3–6, as indicated by the arrow. (**d**) An image of the immunoblotting analysis using antibody against CPB1. A specific band was detected in lanes 3,4,6. (**e**) The peptides splicing sequence after mass spectrometry analysis. O, Oxidation. C, Carbamidomethyl.

**Figure 2 toxins-15-00366-f002:**
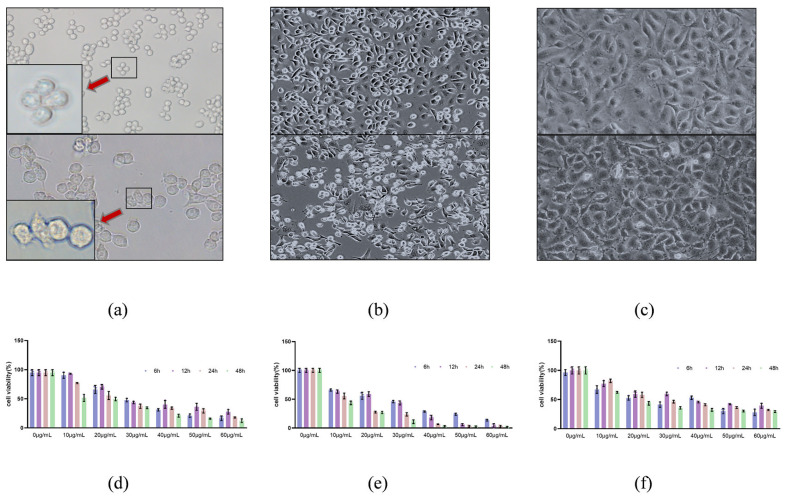
Cytotoxicity of rCPB1 toxin on RAW264.7, NCM460, and HUVEC cells. The cytotoxicity of the rCPB1 protein was ascertained using a CCK-8 assay. (**a**) Image of control RAW264.7 (20×), showing normal morphology of cells, and of cells treated with rCPB1 at the concentration of CT50 for 6 h, which exhibited morphological changes and cell death. Red arrows indicate an enlarged figure (40×). (**b**) NCM460 cells (10×). (**c**) HUVEC cells (10×). (**d**) Viability of RAW264.7 cells. (**e**) NCM460 cells. (**f**) HUVEC cells.

**Figure 3 toxins-15-00366-f003:**
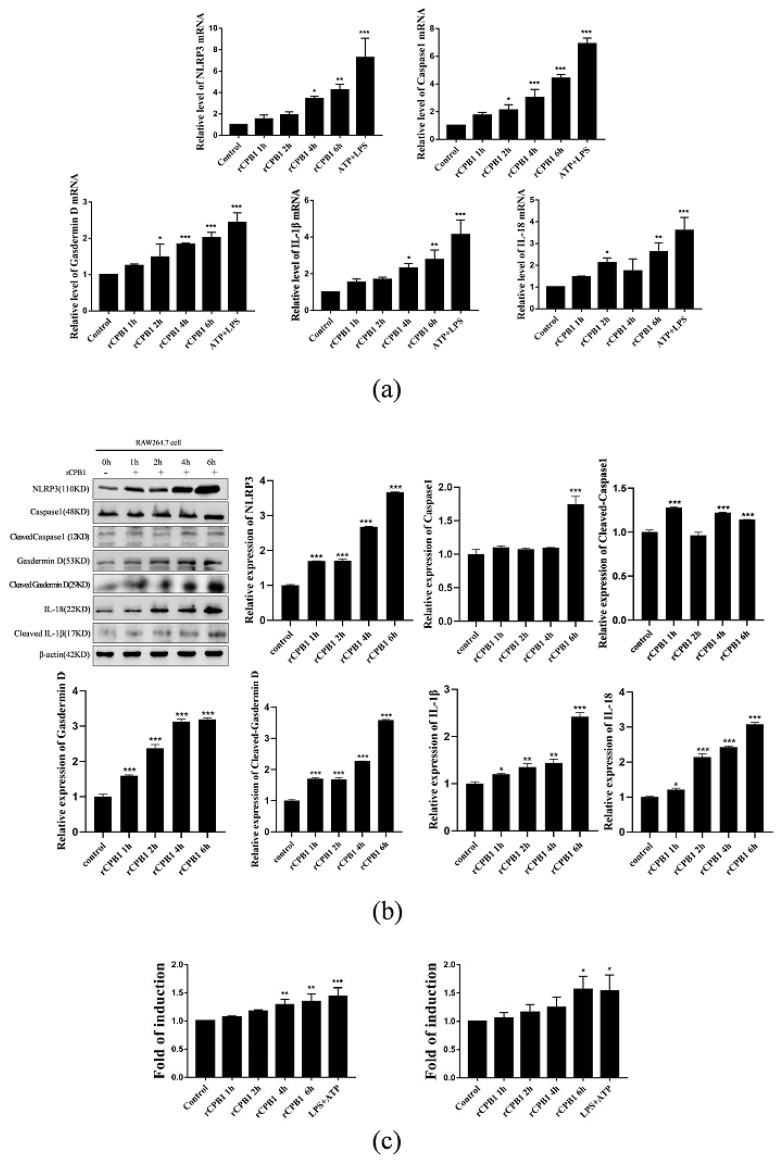
rCPB1 induces pyroptosis in RAW264.7 cells. Cells were treated with rCPB1 at 30 μg/mL for 6 h. (**a**) Image of the expression of pyroptosis-related mRNA in RAW264.7 (**b**) Image of the expression of pyroptosis-related proteins in RAW264.7. (**c**) Expression of IL-1β and IL-18 in the culture supernatants of RAW264.7. *p* < 0.05 *, *p* < 0.01 **, *p* < 0.001 *** all indicated statistically significant differences.

**Figure 4 toxins-15-00366-f004:**
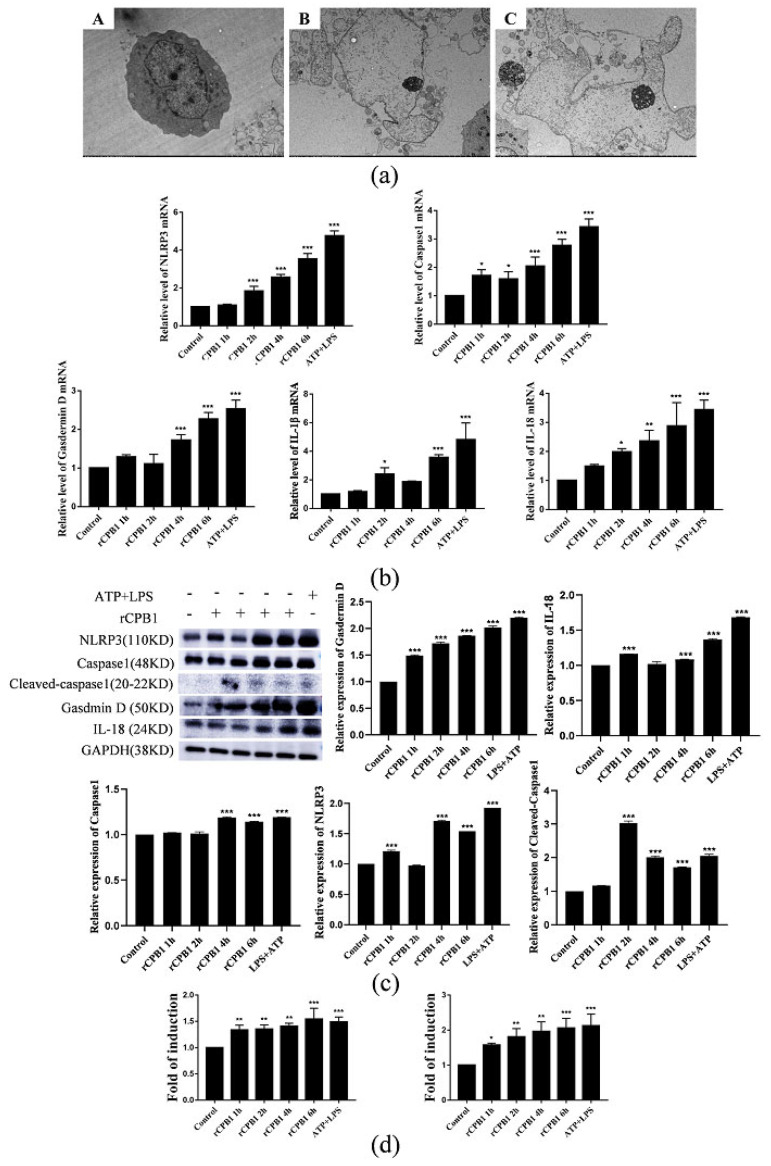
rCPB1 induces pyroptosis in THP−1 cells. Cells treated with rCPB1 at 20 μg/mL for 6 h. (**a**) Image of morphological changes to THP−1 macrophages after rCPB1 treatment under transmission electron microscopy. (**A**): Control group (3000×); (**B**): rCPB1 group (3000×); (**C**): ATP + LPS group (3000×). Red arrows indicate the spillage of cell contents. (**b**) Image of the expression of pyroptosis−related mRNA in THP−1 macrophages. (**c**) Image of the expression of pyroptosis−related proteins in THP−1 macrophages. (**d**) Expression of IL−1β and IL−18 in culture supernatants of THP−1 macrophages. *p* < 0.05 *, *p* < 0.01 **, *p* < 0.001 *** all indicated statistically significant differences.

**Figure 5 toxins-15-00366-f005:**
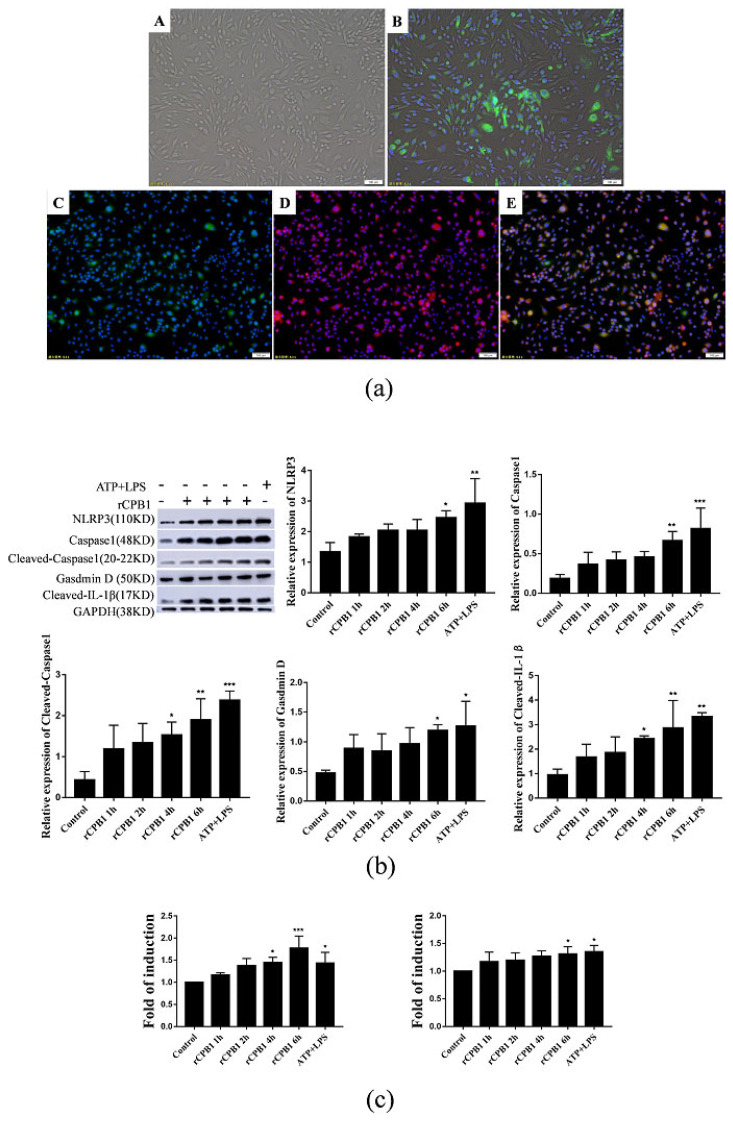
rCPB1 induces pyroptosis in bone marrow−derived macrophages (BMDM). Cells were treated with rCPB1 at 20 μg/mL for 6 h. (**a**) Image of BMDM induced, and the identification of BMDM. (**A**): Cell morphology of BMDM (200×); (**B**): BMDM phagocytized GFP *Escherichia coli* (200×); (**C**–**E**): BMDM immunofluorescence staining with FITC−labeled CD11c, Texas red−labeled CD16, FITC−labeled CD11c and Texas red−labeled CD16 merged (200×). The Chinese in the figure meaning is magnification power. (**b**) Image of the expression of pyroptosis−related proteins in BMDM. (**c**) Expression of IL−1β and IL−18 in culture supernatants of BMDM. *p* < 0.05 *, *p* < 0.01 **, *p* < 0.001 *** all indicated statistically significant differences.

**Figure 6 toxins-15-00366-f006:**
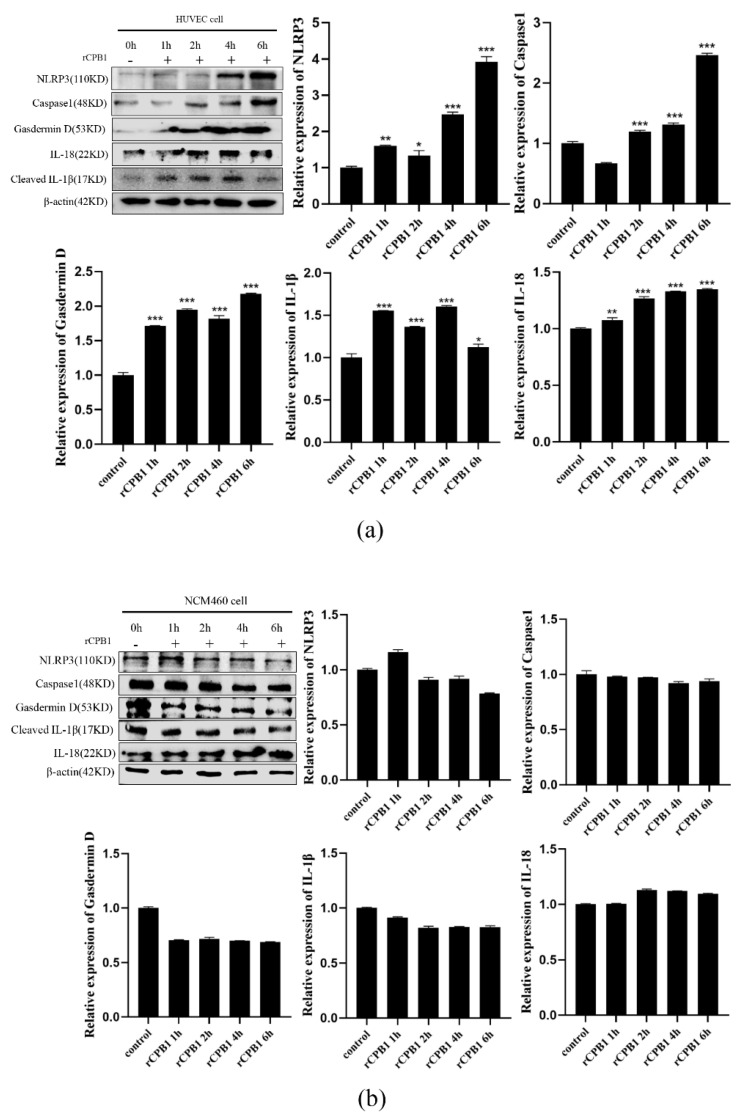
rCPB1 induces pyroptosis in human colon epithelial cells (NCM460) and human umbilical vein endothelial cells (HUVEC). Cells were treated with rCPB1 at 30 μg/mL for 6 h. (**a**) Image of the expression of pyroptosis−related proteins in HUVEC cells. (**b**) Image of the expression of pyroptosis−related proteins in NCM460 cells. *p* < 0.05 *, *p* < 0.01 **, *p* < 0.001 *** all indicated statistically significant differences.

**Figure 7 toxins-15-00366-f007:**
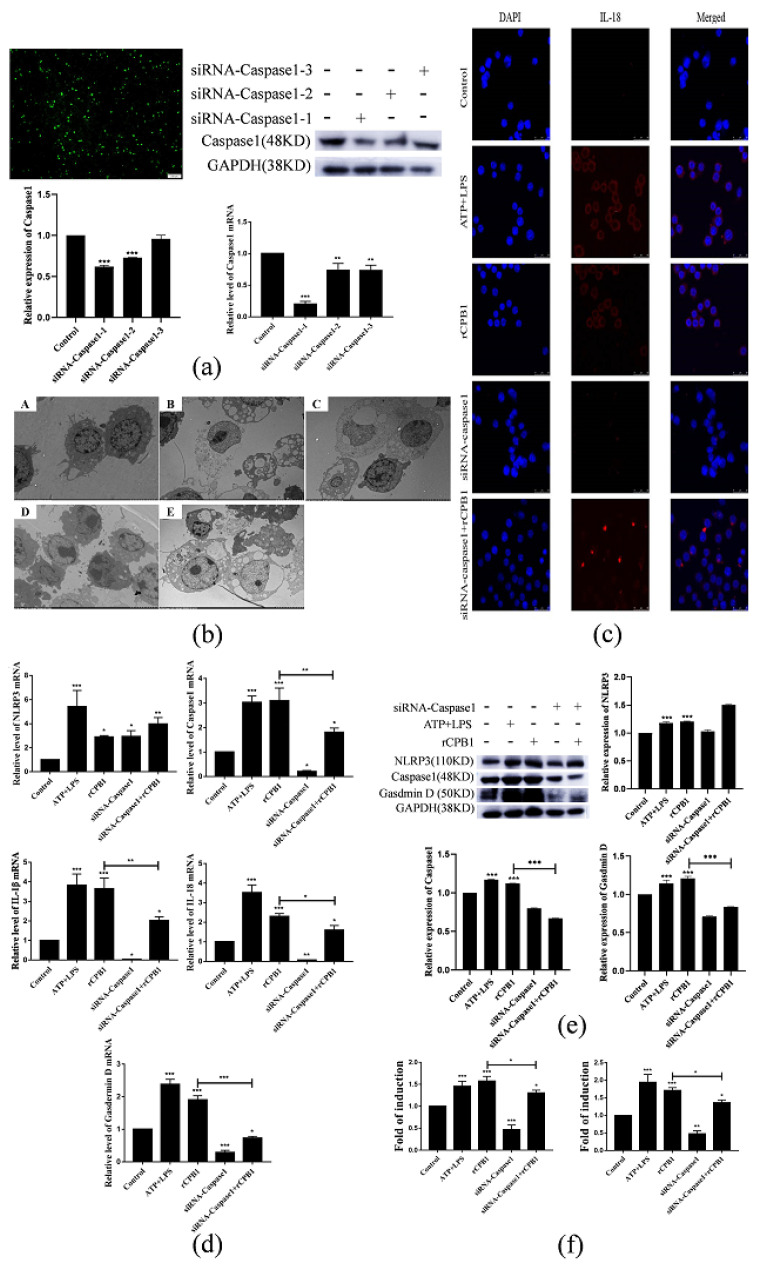
rCPB1 may induced macrophage cell pyroptosis through the typical Caspase−1−dependent pathway. Cells were treated with rCPB1 at 30 μg/mL for 6 h. (**a**) Caspase 1 knock−down in RAW264.7 by small interfering RNA. The mRNA and protein expression results after transfection are shown. (**b**) Transmission electron microscopy image of morphological changes in RAW264.7. (**A**): Control group (2000×); (**B**): rCPB1 group (2000×); (**C**): ATP + LPS group (2000×); (**D**): siRNA (Caspase 1) group (2000×); (**E**): siRNA (Caspase 1) + rCPB1 group (2000×). Red arrows indicate the spillage of cell contents. (**c**) Immunofluorescent staining image of IL−1β in RAW264.7 (400×). (**d**) Image of the expression of pyroptosis−related mRNA in RAW264.7. (**e**) Image of the expression of pyroptosis−related proteins in RAW264.7. (**f**) Image of the expression of IL−1β and IL−18 in the culture supernatants of RAW264.7. *p* < 0.05 *, *p* < 0.01 **, *p* < 0.001 *** all indicated statistically significant differences.

**Figure 8 toxins-15-00366-f008:**
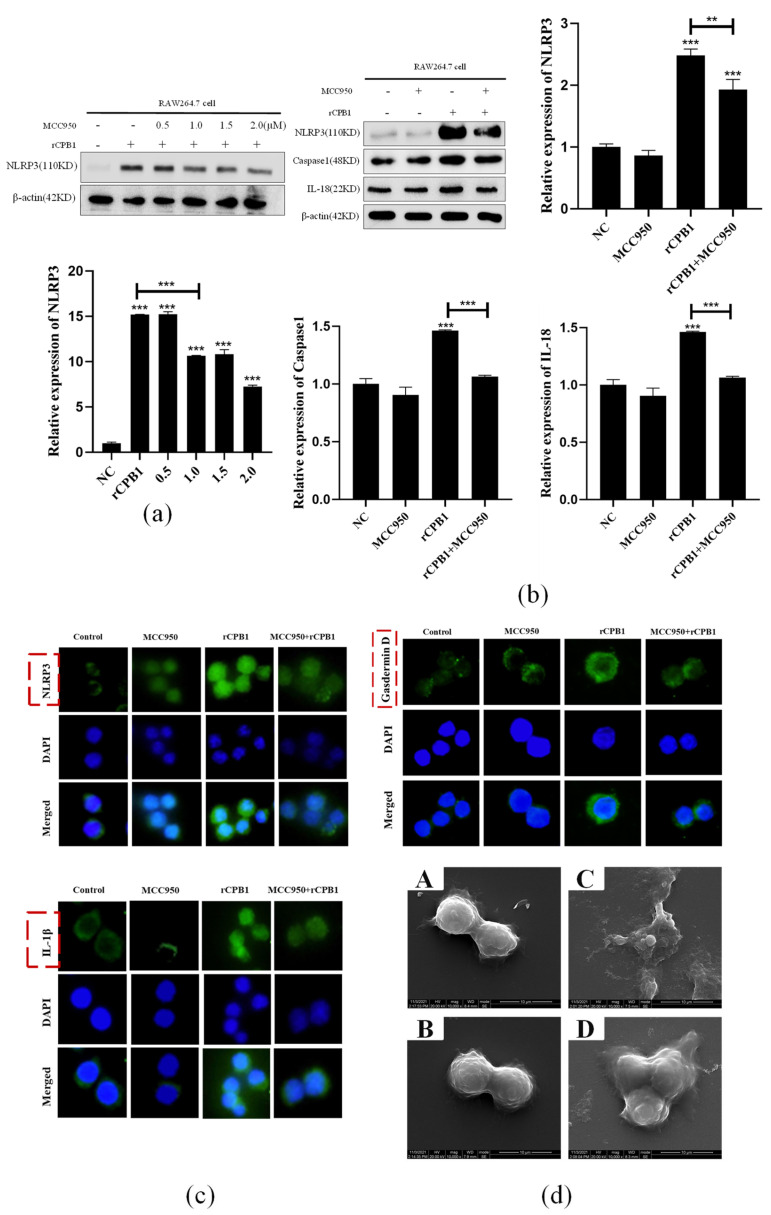
MCC950 significantly inhibited the expression of NLRP3 and its downstream pyroptosis−related proteins. After treatment with MCC950 for 1 h, cells were treated with rCPB1 at 30 μg/mL for 6 h. (**a**) Image of the expression of NLRP3 protein in RAW264.7 cells treated with indicated concentrations of MCC950. (**b**) Image of expression of NLRP3 and its downstream pyroptosis−related proteins in RAW264.7 cells. (**c**) Image of immunofluorescence of NLRP3, Gasdermin D, and IL−1β in RAW264.7 cells (400×). (**d**) Scanning electron microscopy image of morphological changes in RAW264.7 cells after rCPB1 infection and MCC950 inhibition. (**A**): control group (10,000×); (**B**): MCC950 group (10,000×); (**C**): rCPB1 group (10,000×); (**D**): MCC950 + rCPB1 group (10,000×). *p* < 0.05 *, *p* < 0.01 **, *p* < 0.001 *** all indicated statistically significant differences.

## Data Availability

Data is contained within the article or Appendix A.

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
