# Peer review of "Pyroptosis of Macrophages Induced by Clostridium perfringens Beta-1 Toxin"

_toxins, 2023, doi:10.3390/toxins15060366_

Round 1
Reviewer 1 Report
The present manuscript has some significant findings that may have applications in public health and food safety. The study is well-planned and experiments are properly designed. The language is clear and easy to understand except few minor changes required. The hypothesis is clear and sound. The analysis is done with novel and advanced methodologies.
I have following observation in this regard-
Keywords: may be further improved
L24: ang may be and
Introduction: well outlined the background information and justify the need to undertake the study
I would recommend presenting the methodology in a flow chart for quick understanding and increasing the overall impact and citability of the manuscript.
The language is fine and easy to understand
Author Response
Point 1: Keywords: may be further improved.
Response 1: As reviewer suggested that we have added the keywords as “Caspase-1/NLRP3 pathway”.
Point 2: L24: ang may be and.
Response 2: We are very sorry for our typo, and we have made correction “ang” as “and”.
Point 3: I would recommend presenting the methodology in a flow chart for quick understanding and increasing the overall impact and citability of the manuscript.
Response 3: As reviewer suggested that we have presented the methodology in a flow chart.
(Figure S1)
Figure S1: A schematic diagram to summarize our new findings
Special thanks to you for your good comments.

Reviewer 2 Report
The manuscript Pyroptosis of Macrophages Induced by Clostridium perfringens beta-1 toxin describes production CPB1 toxin and its capability to trigger pyroptosis on cell lines of macrophages, colon epithelial cells and macrophages derived from bone marrow.
Labeling of western blots in figure 3, 6 and 8 is in the letters that are too bold and that is why difficult to read.
The evaluation of pyroptosis in figure 3 on RAW264.7, figure 4 on THp-1 cells, figure 5 on bone marrow derived macrophages, then figure 6 on colon epithelial cells. In all these figure are Western done on the same protein, qPCR are done of the primers for the same proteins. There are slight differences. I do not see if gain any significant conclusion from this data, as comparison or generalization.
Author Response
Point 1: Labeling of western blots in figure 3, 6 and 8 is in the letters that are too bold and that is why difficult to read.
Response 1: As reviewer suggested that we have resized (figure 3, 6 and 8).
Point 2: The evaluation of pyroptosis in figure 3 on RAW264.7, figure 4 on THp-1 cells, figure 5 on bone marrow derived macrophages, then figure 6 on colon epithelial cells. In all these figures are Western done on the same protein, qPCR are done of the primers for the same proteins. There are slight differences. I do not see if gain any significant conclusion from this data, as comparison or generalization.
Response 2: We are sorry that you do not see any significant conclusion from this data. Please allow us to answer your questions. RAW264.7 is derived from mouse, and THP-1 is derived from human. Bone marrow derived macrophages are primary cells. NCM460 are epithelial cells, and HUVEC are endothelial cells. We found that whether murine or human macrophages, passing macrophages or primary macrophages, the rCPB1 toxin causes the cells to pyroptosis. The results showed that rCPB1 induced pyroptosis of macrophages. Some studies have found that CPB1 toxin can induced different cytotoxic responses on different cell lines. So, we used different cell lines. We found that the rCPB1 toxin could cause the HUVEC cells released inflammatory factors, indicating that rCPB1 could induce pyroptosis in endothelial cells. However, there was no significant change in the mRNA and protein related pyroptosis which the NCM460 cells treaded by rCPB1 toxin. This result suggests that the rCPB1 toxin protein can cause the death of colonic mucosal epithelium, but does not induce pyroptosis in NCM460. It is speculated that the release of inflammatory factors in animals may be caused by immune cells in the lamina propria of intestinal tissue. Gradually, a large number of inflammatory cytokines are produced through the in-flammatory cascade, which destroys the intestinal epithelium, leading to diseases such as necrotizing enteritis in the host. These are reflected in our discussion.
Special thanks to you for your good comments.
Reviewer 3 Report
Reviewer’s comments
In the current article author has reported the rCPB1 treatment of macrophages promoted the assembly of NLRP3 inflammasomes and activated caspase 1; the activated Caspase 1 caused gasdermin D to form plasma membrane pores, leading to the release of inflammatory factors IL-18 and IL-1β, resulting in macrophage pyroptosis. The current manuscript is written and presented very well. This reviewer is recommending this manuscript for the publication after minor revision.
1. The abstract is not required to subcategorised it can be without legends.
2. The figures quality is not very good. It is quite difficult for reader to follow. The high resolution figures can be include in the revised manuscript.
3. The references are not proper format.
English is good need some minor changes.
Author Response
Point 1: The abstract is not required to subcategorised it can be without legends.
Response 1: As reviewer suggested that we have rewritten abstract.
“Abstract: Clostridium perfringens beta-1 toxin (CPB1) is responsible for necrotizing enteritis and enterotoxemia. However, whether the release of host inflammatory factors caused by CPB1 is re-lated to pyroptosis, an inflammatory form of programmed cell death, has not been reported. A construct expressing recombinant Clostridium perfringens beta-1 toxin (rCPB1) was created, and the cytotoxic activity of the purified rCPB1 toxin was assessed by CCK-8 assay. The rCPB1-induced macrophage pyroptosis was evaluated by accessing changes to the expression of pyroptosis-related signal molecules and the pyroptosis pathway of macrophages using quantitative real-time PCR, immunoblotting, ELISA, immunofluorescence, and electron microscopic Assays. The results showed that the intact rCPB1 protein was purified from an E. coli expression system, which exhibited moderate cytotoxicity on mouse mononuclear macrophage leukemia cells (RAW264.7), normal colon mucosal epithelial cells (NCM460), and human umbilical vein endothelial cells (HUVEC). rCPB1 could induce pyroptosis in macrophages and HUVEC cells, in part, through the caspa-se1-dependent pathway. The rCPB1-induced pyroptosis of RAW264.7 cells could be blocked by inflammasome inhibitor MCC950. These results demonstrated that rCPB1 treatment of macro-phages promoted the assembly of NLRP3 inflammasomes and activated caspase 1; the activated Caspase 1 caused gasdermin D to form plasma membrane pores, leading to the release of in-flammatory factors IL-18 and IL-1β, resulting in macrophage pyroptosis. MCC950 could inhibit rCPB1-induced pyroptosis of RAW264.7 cells as a therapeutic target. This study provided a novel insight into the pathogenesis of CPB1.”
Point 2: The figures quality is not very good. It is quite difficult for reader to follow. The high resolution figures can be include in the revised manuscript.
Response 2: It is really true as reviewer suggested that the figures quality is not very good. And we have replaced high resolution figures in the revised manuscript (all figures).
Point 3: The references are not proper format.
Response 3: As reviewer suggested that we have made correction.
Special thanks to you for your good comments.